# Discovering the Physio-Pathological Mechanisms of Interaction between Bone Mineral Density, Muscle Mass, and Visceral Adipose Tissue in Female Older Adults through Structural Equation Modeling

**DOI:** 10.3390/jcm12062269

**Published:** 2023-03-15

**Authors:** Simone Perna, Clara Gasparri, Sabika Allehdan, Antonella Riva, Giovanna Petrangolini, Cinzia Ferraris, Davide Guido, Tariq A. Alalwan, Mariangela Rondanelli

**Affiliations:** 1Division of Human Nutrition, Department of Food, Environmental and Nutritional Sciences (DeFENS), Università degli Studi di Milano, 20122 Milano, Italy; 2Endocrinology and Nutrition Unit, Azienda di Servizi alla Persona ‘‘Istituto Santa Margherita’’, University of Pavia, 27100 Pavia, Italy; 3Department of Biology, College of Science, University of Bahrain, Sakhir Campus, Zallaq P.O. Box 32038, Bahrain; 4Development Department, Indena SpA, 20139 Milan, Italy; 5Food Education and Sport Nutrition Laboratory, Department of Public Health, Experimental and Forensic Medicine, University of Pavia, 27100 Pavia, Italy; 6Independent Researcher, 00168 Rome, Italy; 7IRCCS Mondino Foundation, 27100 Pavia, Italy; 8Unit of Human and Clinical Nutrition, Department of Public Health, Experimental and Forensic Medicine, University of Pavia, 27100 Pavia, Italy

**Keywords:** sarcopenia, osteoporosis, visceral obesity, metabolic profile, vitamin D

## Abstract

This study aims to examine the relation between visceral adipose tissue (VAT), as a proxy for metabolically unhealthy obesity, muscle, as a proxy for muscle quality and sarcopenia, and bone, as a proxy for bone mineral density and osteoporosis. Other variables, such metabolic syndrome, nutritional status, number of diseases, kidney and liver function and inflammation were assessed as direct or indirect effects. This study used structural equation modeling (SEM) in a sample of 713 older women (mean age 82.1 ± 6.3). The results indicate a positive statistically significant association between bone and muscle mass (β = 0.195, <0.001) and nutritional status and muscle mass (β = 0.139, *p* < 0.001), but negative association between age with muscle mass (β = −0.509, *p* < 0.001) and nutritional status (estimates: −2.264, *p* < 0.001). A negative association between VAT and muscle mass was also reported (β = −1.88, *p* < 0.001). A negative statistically significant association was reported between bone mineral density and functional status (β = −1.081, *p* < 0.001), and a positive association between functional status and muscle mass (β = 9.000, *p* < 0.001). In addition, functional status was positively statistically associated with cognitive performance (β = 0.032, *p* < 0.001). The SEM method demonstrates that the VAT, muscle mass and bone mineral density are associated, but the form of the relation is different in relation to different factors, such as nutritional status, mental and functional status, age, and number of pathologies, having different impacts on metabolic outcomes. SEM is a feasible technique for understanding the complex mechanisms of frailty in the elderly.

## 1. Introduction

Studying the mechanisms of interaction of visceral adipose tissue (VAT), muscle mass and bone mineral density is very important because they represent major factors of frailty in the elderly and are associated with numerous health complications [1].

A consensus has emerged that fat stored in the central segment of the body is particularly damaging, in that it portends greater risk for frailty and consequences linked to metabolic syndrome such as diabetes, cardiovascular disease, hypertension, and certain cancers [2].

However, there is less information regarding the mechanisms that may link visceral fat with risk for of sarcopenia and osteoporosis, since low bone mineral density in the elderly leads increased mortality and muscle complications and vice versa [3]. 

Both diseases share common health complications, but there are inconsistent findings concerning the relationship between visceral obesity and muscle [4]. 

In particular, VAT could affect the bone cycle through different mechanisms; for example, there is evidence that vitamin D metabolism, storage, and action both influence and are influenced by adiposity. An increased risk of vitamin D deficiency in those people with obesity may influence the mobilization of free fatty acids from the adipose tissue [5].

In addition, the link between VAT, metabolic syndrome and consequent derangement of muscle and bone is clear: the expression of enzymes related to lipid turnover in visceral fat (e.g., lipoprotein lipase, hormone-sensitive lipase, peroxisome proliferator-activated receptor) increase with fat feeding in visceral fat [6].

As reported by Huo, the combination of osteoporosis and sarcopenia has a clear linkage, and nutritional status seems to influence this relationship. In particular, the mechanism linking nutritional markers such as serum folate to muscle and bone health could involve direct effects on muscle cells, as it is essential for intracellular processes including prevention of DNA damage, lowering of oxidative stress, and inhibition of apoptosis [7].

Given this background, the aim of this study was to use structural equation modeling (SEM) in order to understand the following mechanisms (1) if muscle mass (assessed by appendicular lean mass divided by height square ALM/h2, as well as handgrip strength) and bone mineral density, assessed via DXA, are directly and positively associated; (2) if muscle mass and bone mineral density are negatively affected by VAT; (3) if muscle mass is directly influenced by nutrition (evaluated by mini nutritional assessment, albumin and vitamin D) and number of diseases; (4) if muscle mass affects functional status (assessed via Barthel index and activities of daily living); (5) if VAT negatively affects metabolic profile (assessed via glycemia, lipid profile, waist circumference), inflammation (assessed using CRP) and liver profile.

## 2. Materials and Methods

### 2.1. Study Design and Population

Data collection included Italian hospitalized elderly (in Santa Margherita Institute, Azienda di Servizi alla Persona di Pavia, Department of Public Health, University of Pavia, Italy). This is a cross-sectional study with the application of structural equation modeling. 

### 2.2. Inclusion and Exclusion Criteria

The inclusion criteria were: (1) admission to the postacute geriatric care unit for functional loss secondary to a non-disabling medical disease, (2) aged 65 years or older, (3) willingness to participate and provide a signed informed consent. At time of admission, the patients were not diagnosed with disabling diseases that could directly affect muscle weakness (such as neurological diseases, hip fractures or amputations). Exclusion criteria were: subjects affected by acute illness, severe liver (as defined by ESPEN Guidelines) [8], heart (European Society of Cardiology proposed guidelines for the diagnosis) [9] or kidney dysfunction (acute kidney “risk, injury, failure” as defined by the newly developed RIFLE classification) [10] or severe dementia (MMSE < 18 points) [10]. Participants with diabetes, metabolic disease or neoplasia, as well as patients treated with steroids or those who were unable to walk were excluded. In addition, BMD and T score of femur neck, BMD and T score of lumbar spine (L1–L4), and risk of fractures at 10 years (FRAX) of major osteoporotic fractures were not collected because of high risk of bias due to spinal compression in the elderly. Additionally, the bone turnover markers available were not included in the study protocol. 

### 2.3. Outcomes

The data were collected in a 10-year period from January 2011 to January of 2021 in collaboration with the University of Pavia. The study design was approved by the ethics committee of the University of Pavia and an individual written informed consent was obtained from each participant. 

The following variables have been suggested in the literature to be associated with the relation between visceral fat, bone and muscle. This study considered the following observed, endogenous variables: hip T-score, hip FRAX, ADL, Barthel score, VAT, android fat %, handgrip test, MNA, waist circumference, triglyceride glycemia, CRP, creatinine, blood urea nitrogen (BUN), AST, ALT, GGT and albumin in grams. The observed, exogenous variables were MMSE, number of diseases, vitamin D and age. The unobserved, endogenous variables were the functional status, the metabolic profile, kidney function and liver function.

### 2.4. Anthropometric Parameters and Body Composition

Body weight and height were measured following a standardized technique [11] and the body mass index (BMI) was calculated (kg/m^2^). Anthropometric parameters were always collected by the same investigator. 

Body composition (fat free mass (FFM), fat mass (FM), and gynoid and android fat distribution) was measured by dual-energy X-ray absorptiometry (DXA) with the use of a Lunar Prodigy DXA (GE Medical Systems). The in vivo CVs were 0.89% and 0.48% for whole body fat (FM) and FFM, respectively. 

Visceral adipose tissue volume was estimated using a constant correction factor (0.94 g/cm^3^). The software automatically placed a quadrilateral box, representing the android region, outlined by the iliac crest and with a superior height equivalent to 20% of the distance from the top of the iliac crest to the base of the skull [12]. The Lunar Prodigy DXA (GE Medical Systems) was used to measure T-scores of the hip and femur. The skeletal muscle index was taken as the sum of the fat-free soft tissue mass of arms and legs divided by height ^2^ [13].

### 2.5. Anthropometric Parameters and Body Composition

The Mini Nutritional Assessment (MNA) was used to assess the nutritional status of all participants [14]. This tool uses measurements and a short questionnaire, including an anthropometric assessment of weight, height, and weight loss, in addition to a general assessment (e.g., lifestyle, medication, and mobility), and a dietary assessment (i.e., number of meals, food and fluid intake, self-assessment of autonomy of eating, and self-perception of health and nutrition). The waist was measured at the midpoint between the top of the hip bone (iliac crest) and lowest rib using a standardized method. 

### 2.6. Muscle Function

Muscle function was assessed using a Jamar hand dynamometer, handgrip, (Jamar 5030J1; Sammons Preston Rolyan; accuracy 0.6 N), by means of a standardized procedure [15]. 

### 2.7. Cognitive Performance and Functional Status 

The Mini Mental State Examination (MMSE) was administered in order to evaluate cognitive status [16]. To evaluate independent living, the participants were tested using the Barthel index (BI) [17], a simple index to score the ability of a patient with a disability disorder related to functional status to care for themself. The participants’ ability to care for themselves was assessed with the Katz Index of Independence in Activities of Daily Living [18].

### 2.8. Biochemical Analysis

Blood samples were collected; in particular, nutritional status, lipid profile, glycemic profile and status of inflammation were assessed. Lipids and creatinine were measured via enzymatic-colorimetric assay (Abbott Laboratories). CRP, was determined via immunoturbidimetry (Roche). Insulin was measured via electro-chemiluminescence immuno-assay (ECLIA) (Roche Diagnostics). Blood glucose, aspartate aminotransferase (AST) and alanine aminotransferase (ALT) were analyzed via Enzymatic UV Assay (Abbott Laboratories). VitD25OH was determined via ECLIA (Roche). 

### 2.9. Prespecified Structural Equation Modeling (SEM) of the Visceral Obesity–Muscle Mass–Bone Mineral Density Interactions

Structural equation modeling is based on the analysis of both observed and non-observed (i.e., latent) variables or constructs. It is a multivariate approach based on the use of a system of simultaneous equations to describe a priori path relationships that generate the data, where a given variable can appear explanatory in one or several equations as well as being the outcome in other equations. 

The analysis uses a combination of correlation measures and regressions to assess direct and indirect pathways within a pre-specified pathophysiological mode. Figure 1 shows the pre-specified structural equation modeling for understanding the interconnections between visceral obesity, muscle and bone.

The proposed model (see Figure 1) was tested using SEM and AMOS 5.0 software. SEM is a useful statistical procedure for researchers who want to test a theory involving causal processes, and therefore is well suited to the management of cross-sectional data for inferential purposes. With regard to the conceptual model of physiopathological interactions, the observed variables load on the latent variables (factors) in the Figure 1 pattern. Kirk et al. suggested a model in which “bone-fat-muscle” affects directly and indirectly the “metabolic factors” [19]. Although this model does not completely explain the physiological pathways between the diseases of these three factors, it has been used as an empirical framework in studies of physiology. This model consists of three main unobserved latent variables of muscle (for sarcopenia), visceral fat (for visceral fat markers) and bone (t score, risk of fractures FRAX) and eight observed variables (Figure 1) covariated with themselves. 

### 2.10. Statistical Analysis

Descriptive analysis: Continuous variables were summarized as the mean standard deviation (SD) or the median [interquartile range (IQR)]. Structural equation modeling is a method for representing, estimating and testing a theoretical network of mostly linear relations between variables that may be either directly observable or unobservable and may only be measured imperfectly. 

The general SEM model can be decomposed into two sub-models: a measurement model and a structural model. The measurement model defines relations between the observed and unobserved latent variables. The structural model defines relations among the unobserved variables by specifying the pattern by which particular latent variables directly or indirectly influence some other latent variables in the model.

SEM is mainly a confirmatory technique rather than exploratory and is more likely to be used to determine whether a certain model is valid rather than to find a suitable model. However, SEM analysis often involves a certain degree of exploratory analysis. By convention, when graphically representing the model, the observed variables are enclosed by rectangles or squares and latent variables are enclosed by ovals or circles. Residuals are always unobserved and are represented by ovals or circles. This analysis was performed using AMOS software [20].

Primary Variables (covariates between them):(1)Muscle mass (latent variable) includes ALM/h2 and handgrip strength;(2)Visceral fat (latent variable) includes VAT (grams) and android fat (%);(3)Bone mineral density (latent variable) includes T-score for hip and femur and hip FRAX (%).

Secondary Variables 

(1)Nutritional status (latent variable) includes Mini Nutritional Assessment and albumin and vitamin D (single variable);(2)Liver status (latent variable) includes ALT, GGT, ALP;(3)Functional status (latent variable) includes Barthel test and ADL;(4)Cognitive performance includes MMSE;(5)Age, gender;(6)Kidney function (latent variable) includes creatinine and azotemia (blood urea nitrogen) (BUN));(7)Metabolic profile (latent variable) includes triglycerides, glycemia, and waist circumference.

This hypothesized model is a non-recursive model, that is, it is a model with three structural equations where the dependent variable of each equation appears as a predictor variable in the other equation. In this model, visceral fat, muscle and bone form a feedback loop, meaning that we can follow the path between these three variables an infinite number of times without having to return to the other variables. This model states that muscle mass is directly influenced by visceral fat and vice versa. Similarly, bone is directly influenced by visceral fat and vice versa.

## 3. Results

A summary of descriptive statistics is given in Table 1. As this table shows, the 713 females had a mean age of 82.13 ± 63 years old with a BMI of 24.99 ± 5 kg/m^2^. The sample was in a situation of osteopenia or osteoporosis according to femur t-score (−2.39 ± 1.22 ds), with high risk of femur fracture based on FRAX (8.61%). In overall, the total sample was in a situation of sarcopenia or pre-sarcopenia according to SMI and a condition of mild malnutrition (mean MNA was 17.47 points). Vitamin D was deficient or insufficient (13.98) and all other biochemical parameters were in the normal range.

Figure 2 reports the standardized path coefficients (β values) and estimates of the raw path coefficients. The *p* values in Table 2 were achieved by Wald’s z test on the raw coefficients.

A strong positive statistically significant association was recorded between bone mineral density and muscle mass with standardized path coefficient of β: +0.70). A negative association between visceral fat with muscle mass ((β: −0.14) and visceral fat with bone mineral density (β: −0.01) was found.

A pure direct effect of “nutritional status” on “muscle” was observed (β: +0.31). Nevertheless, the interaction “visceral fat” × “nutrition” (β: −0.01) was not statistically significant. “Age” was found to be associated negatively with bone (β: −0.31) and negatively with muscle mass (β: −1.00).

Table 3 shows the raw path coefficients of the selected fitted model from the additional testing strategy for interaction effects between explanatory variables in the equations of the outcome variables. A statistically significant association was reported between bone and muscle mass (estimates: 0.195, *p* < 0.001), nutrition and muscle mass (estimates: 0.139, *p* < 0.001) and age with muscle mass (estimates: −0.509, *p* < 0.001) and nutritional status (estimates: −2.264, *p* < 0.001). A negative association between visceral fat and muscle mass was also reported (estimates: −1.88, *p* = 0.083).

Table 4 shows the raw path coefficients of the selected fitted model from the additional testing strategy for interaction effects between explanatory variables in the equations of the outcome variables. A negative statistically significant association was reported between bone mineral density and functional status (estimates: −1.081, *p* < 0.001) and a positive association with muscle (estimates: 9.000, *p* < 0.001) was found. In addition, functional status was statistically associated with mental status (estimates: 0.032, *p* < 0.001).

## 4. Discussion

To the best of our knowledge, this is the first study performed in a female aging population to evaluate the mechanisms of interaction between VAT, muscle mass and bone mineral density and, additionally, understand the role of nutritional status, functional status, metabolic profile, cognitive performance and age in this complex relationship. 

### 4.1. The Effect of VAT on Muscular Status and Strenght 

Primarily, our study demonstrates that muscular status is influenced negatively by the VAT. As demonstrated by Alalwan et al. in 2020, the accumulation of adipose tissue or the presence of adipocyte-infiltrating macrophages leads to increased secretion of proinflammatory cytokines, such as interleukin (IL)-1, IL-6, and tumor necrosis factor-α (TNF-α) that affects muscle growth and regeneration [21]. 

Li et al. highlighted that during aging, adipose inflammation leads to the redistribution of fat to the intra-abdominal area (visceral fat) and fatty infiltrations in skeletal muscles, resulting in decreased overall strength and functionality and enhancing reactive oxygen species (ROS) production, leading to lipotoxicity and insulin resistance, as well as enhanced secretion of some pro-inflammatory cytokines [22]. 

Regarding the interaction between bone and VAT, the mechanism of the negative relationship between is multifaceted and with several controversies. 

As found by Rondanelli et al., the unmatching between fat and bone mineral density could be explained because of biomechanical forces or increased aromatization of androgens to weak estrogens in subcutaneous fat tissue, and in women, higher peripheral aromatization of estrogen is associated with adiposity [1].

### 4.2. The Nutritional Status’ Mediation between Visceral Fat and Muscle Mass

Another important factor is nutritional status, which could mediate the negative effects of visceral fat on muscle mass, affecting both on different sides.

As demonstrated by Chang et al., the interaction effects between central obesity and sarcopenia on nutritional status might reflect the dilemma of the obesity paradox when treating older people [23]. Indeed, except the MNA, central obesity and sarcopenia might also be an important surrogate for assessing the nutritional status of older people. To the best of our knowledge, this is the first study investigating the relationship between VAT and nutritional status and muscle. 

As demonstrated by Bartosz et al., there is some evidence that malnutrition is linked to the development of visceral adiposity [24]. 

### 4.3. Muscle Deterioriation with Age and Comorbidities 

Moreover, this study found that age and multiple diseases, in combination with VAT, have a primary effect on muscle deterioration. In particular, peri-muscular fat in older age could further exacerbate age-related muscular atrophy as examined by Morrison et al. in ectopic fat accumulation layered around atrophied hindlimb skeletal muscle [25].

Regarding the interaction between visceral fat, age and muscle mass, a recent study by Larsson found that histochemical changes in muscle tissue such as decreased proportion of type II fibers and selective atrophy of type II fibers were seen with increasing age, and a decline in strength in old age was also observed to correlate significantly with the type II fiber area [26]. 

### 4.4. Interaction between Muscle and Bone

This study shows that there is a clear positive association between muscle mass and bone mineral density. As found by Rondanelli et al. in 2014 and by Kaij, muscle–bone relationships include two factors: local control of muscle to bone and systemic humoral interactions between muscle and bone where osteoglycin might be one of the muscle-derived humoral bone anabolic factors [1,27]. Another possible explanation has been suggested by Ilich at al, postulating that muscle is the dominant positive predictor of femoral neck, total femur, and spine bone mineral density in normal-weight to slightly overweight postmenopausal women [28].

### 4.5. The Effects of Cognitive Status on Muscle Deterioriation 

Furthermore, cognitive performance decline was associated with functional status and indirectly associated with bone and muscle status. 

Behind nutritional status, functional status also has an important impact on muscle quality, as demonstrated in our study. We found an important association between mental status and functional status and a direct association between functional status and muscle. As shown by Foldvari et al. [29], leg power and muscle quality in the legs is a strong predictor of self-reported functional status in elderly women. 

In addition, there is an important effect of the cognitive status on functional status that indirectly plays a role in muscle quality. As found by Lin et al., there is an important relation between mood state and muscle strength at discharge and not just early rehabilitation alone [30]. Finally, the number of pathologies also affects muscle mass, since many chronic diseases lead to chronic inflammation, poor functionality and cognitive decline. Specific patients with chronic diseases such as T2DM patients with high visceral fat accumulation have low muscle quality, as demonstrated by Murai et al. [20].

### 4.6. Limitations of the Study

Regarding limitations, and as direction for further studies, we think that the multidimensional role and effects of subcutaneous fat should be considered, and we did not consider these aspects in our study. In this context, nutritional status was positively associated with muscle and negatively associated with visceral adipose tissue. As an additional limitation, this study was performed only on women and the data are not generalizable to all elderly population, especially since our study involved a cohort of women from only one hospital. The number of drugs in a patient’s regimen and their previous hospitalizations were not investigated. As an additional limitation, this study did not consider the BMD and T score of femur neck, BMD and T score of lumbar spine (L1–L4), and risk of fractures at 10 years (FRAX) and bone turnover markers. These data were not considered because of the high prevalence of spine compression in our sample.

Moreover, it has recently been demonstrated that Denosumab demonstrates positive impact and significant improvements both in BMD and sarcopenia measures [31]. These results were generalized to a population of elderly population with mean of age 82 years old admitted to a geriatric rehabilitation unit. The population covered normal and overweight subjects with multiple conditions such as sarcopenia, osteoporosis, malnutrition, and diabetes.

## 5. Conclusions

Our SEM method demonstrated interesting physiological effects and interactions of the three major causes of frailty, where it seems that visceral fat is a major indicator of frailty and bone mineral density is negatively influenced by visceral adipose tissue, playing a crucial and pivotal role in the frailty profile of elderly women. Different factors, such as nutritional status, mental and functional status, age, number of pathologies having different impact on muscle and bone, were found to have direct or indirect relationships to frailty. Our study also demonstrated that SEM is a feasible technique for understanding the complex mechanisms of frailty in the elderly.

## Figures and Tables

**Figure 1 jcm-12-02269-f001:**
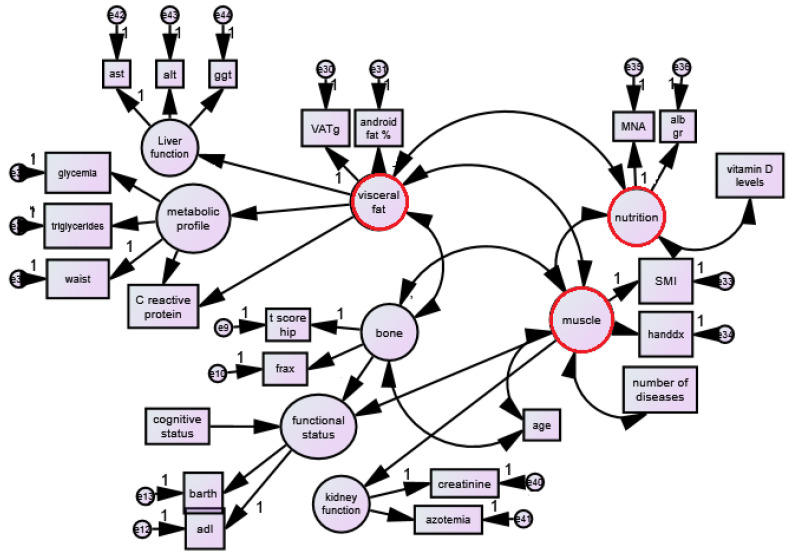
The path model of interaction of bone mineral density, muscle mass and visceral fat with all other factors.

**Figure 2 jcm-12-02269-f002:**
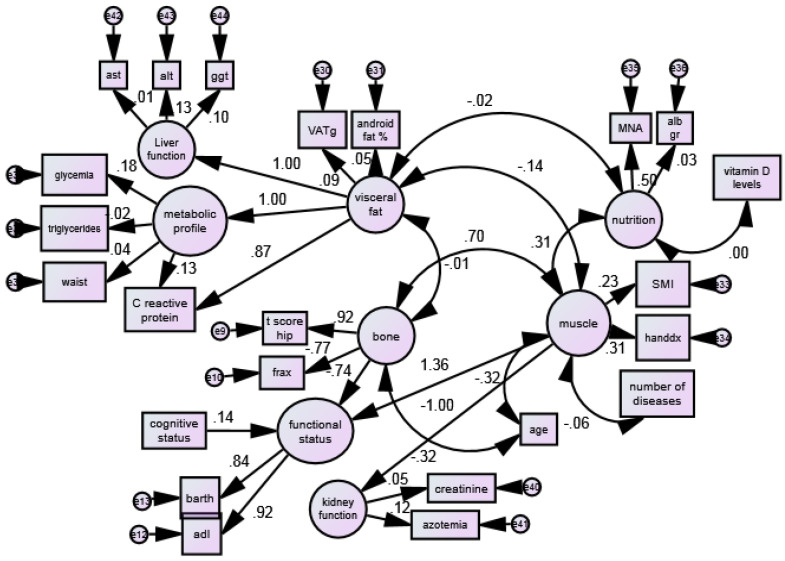
SEM model testing the association between muscle, bone, visceral fat and other functional–metabolic factors in a sample of 713 older adult women.

**Table 1 jcm-12-02269-t001:** Means and standard deviations for all variables used in the analysis.

Variable	Mean	Std. Deviation
Age (years)	82.13	6.32
BMI (kg/m^2^)	24.99	5.00
Mini Mental State Examination (score)	18.13	7.02
Barthel test (score)	61.03	25.93
Adl (score)	3.13	1.81
Iron (mg/dL)	65.60	31.27
Triglycerides (mg/dL)	123.07	59.33
Cholesterol (mg/dL)	190.10	44.52
Albumin (g)	3.96	4.19
Creatinine (mg/dL)	0.93	2.09
Blood urea nitrogen (BUN) (mg/dL)	43.96	20.21
AST (IU/L)	19.72	12.89
ALT (IU/L)	17.31	14.62
GGT (U/L)	31.46	38.52
Glycemia (mg/dL)	106.72	39.96
CRP (mg/dL)	1.11	2.38
25 OH vitamin D (ng/mL)	13.98	11.87
Waist circumference (cm)	90.02	12.12
MNA (score)	17.47	3.58
Handgrip dx (kg)	14.80	5.43
SMI (Kg/m^2^)	6.37	1.09
Femur T score (pt)	−2.39	1.22
Hip FRAX (%)	8.61	7.80
Android fat (%)	36.90	13.55
VAT (g)	911.62	607.33

CRP: C reactive protein, MNA: Mini Nutritional assessment, VAT: Visceral Adipose Tissue, SMI: Skeletal Muscle Index, AST: Aspartate transaminase or aspartate, GGT: Gamma-glutamyltransferase, ALT: Alanine transaminase, 25 OH vitamin D: 25-hydroxycholecalciferol.

**Table 2 jcm-12-02269-t002:** Factor loadings (variable–factor estimates) and CFA indices of latent variable measurement models.

Variable			Estimate	S.E.	C.R.	*p*
**BONE**						
tscorefemore	<---	Bone	1.000			
fraxanca	<---	Bone	−5.287	0.561	−9.415	**<0.001**
**FUNCTIONAL STATUS**						
ADL (score)	<---	Functional status	1.000			
Barthel test (score)	<---	Functional status	13.143	0.971	13.538	**<0.001**
**VISCERAL FAT**						
VAT (grams)	<---	Visceral fat	1.000			
Android fat (%)	<---	Visceral fat	0.013	0.012	1.098	**<0.001**
**MUSCLE**						
SMI (kg/m^2^)	<---	Muscle	1.000			
Handgrip (kg)	<---	Muscle	6.716	1.576	4.261	**<0.001**
**NUTRITIONAL STATUS**						
MNA (score)	<---	Nutrition	1.000			
Albumin gr	<---	Nutrition	0.062	0.232	0.267	0.790
**METABOLIC LE PROFILE**						
Waist circ.(cm)	<---	Metabolic profile	1.000			
Triglyceride (mg/dL)	<---	Metabolic profile	−1.971	5.506	−0.358	0.720
Glycemia (mg/dL)	<---	Metabolic profile	14.446	17.154	0.842	0.400
**KIDNEY FUNCTION**						
Creatinine (mg/dL)	<---	Kidney function	1.000			
Blood urea nitrogen (BUN) (mg/dL)	<---	Kidney function	23.651	21.744	1.088	0.277
**LIVER FUNCTION**						
AST (mg/dL)	<---	Liver function	1.000			
ALT (mg/dL)	<---	Liver function	11.201	33.260	0.337	0.736
GGT (mg/dL)	<---	Liver function	21.632	64.608	0.335	0.738

SMI: skeletal muscle index, MNA: Mini Nutritional Assessment.

**Table 3 jcm-12-02269-t003:** Main model—Covariates path coefficients estimated using regression (linear) in a structural equation model of the relationship between bone mineral density, muscle mass, visceral fat components, nutritional status, vitamin D, no. of diseases and age.

Variable			Estimate	S.E.	C.R.	*p*
Bone	<-->	Visceral fat	−0.541	2.487	−0.218	0.828
Bone	<-->	Muscle	0.195	0.044	4.466	**<0.001**
Visceral fat	<-->	Muscle	−1.889	1.090	−1.733	0.083 *
Nutrition	<-->	Muscle	0.139	0.042	3.280	**<0.001**
Nutrition	<-->	Visceral fat	−1.684	8.167	−0.206	0.837
N’ of diseases	<-->	Muscle	−0.046	0.024	−1.897	0.058
25 OH vitamin D	<-->	Nutrition	−0.032	2.363	−0.014	0.989
Age (years)	<-->	Muscle	−0.509	0.116	−4.388	**<0.001**
Age (years)	<-->	Bone	−2.264	0.313	−7.226	**<0.001**

Bold: *p* value < 0.001. *: suggested values close to the *p* value < 0.05.

**Table 4 jcm-12-02269-t004:** Path coefficients estimated using regression (linear or logistic) in a structural equation model of the relationship between frailty components, mortality and morbidity in older outpatients with cancer.

			Estimate	S.E.	C.R.	*p*
Functional status	<---	Bone	−1.081	0.340	−3.182	**<0.001**
Functional status	<---	Muscle	9.000	1.985	4.534	**<0.001**
Functional status	<---	MMSE	0.032	0.010	3.022	**<0.001**
Metabolic profile	<---	Visceral fat	0.010	0.012	0.797	0.425
Kidney function	<---	Muscle	−0.422	0.368	−1.147	0.251
Liver function	<---	Visceral fat	0.003	0.010	0.334	0.738

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
