# Peer review of "Discovering the Physio-Pathological Mechanisms of Interaction between Bone Mineral Density, Muscle Mass, and Visceral Adipose Tissue in Female Older Adults through Structural Equation Modeling"

_jcm, 2023, doi:10.3390/jcm12062269_

Round 1

Reviewer 1 Report

It is a very interesting article for understanding the complex mechanisms of frailty in elderly.

The "results" part of the manuscript is based only on tables and figures. I suggest adding explanatory text.

Table 1: units for iron are missing

There are a few typos to correct:

Table 2: tscorefemore --> t-score femur; fraxanca --> hip fracture; handrip --> Hand Grip; triglycerides; azototemia --> azotemia; capital letters for: AST; ALT and GGT.

Even if English is not my mother tongue, I suggest you proofread the article to improve the English used.

Author Response

To Editor,

We revised the manuscript with modifications and changes based on the new reviewer’s comments and editor’s comment (pdf notes).

We send you the revised manuscript together with our point-by-point response.

The changes in the text are written in red.

Thanking you in advance for your kind collaboration and suggestions.

Best regards,

The authors

Reviewer 1

The "results" part of the manuscript is based only on tables and figures. I suggest adding explanatory text.

Table 1: units for iron is missing

A: Thanks a lot for your suggestion. Unit of measure for iron has been added and more explanatory text has been included into the manuscript.

Table 2: tscorefemore --> t-score femur; fraxanca --> hip fracture; handrip --> Hand Grip; triglycerides; azototemia --> azotemia; capital letters for: AST; ALT and GGT.

 A: A: Thanks a lot for your suggestion, we modified all these outcomes accordingly

Reviewer 2 Report

This study analyzes and describes the physio-pathological interaction between bone mineral density, muscle mass and visceral adipose tissue (VAT) in old women using the structural equation modeling (SEM). This article is certainly original and interesting, in the light also of the recent literature on the metabolic effects of VAT. However, in my opinion, the analysis relating to bone tissue should be implemented. In the study, only "T score" femur and risk of hip fracture (FRAX) are described. I think that, the authors could complete the analysis by also adding: BMD and T score of femur neck, BMD and T score of lumbar spine (L1-L4), and risk of fractures at 10 years (FRAX) alo of major osteoporotic fractures. Moreover, are Bone turnover markers available? could they be associated to VAT? In table 1 and in the text: "vitamin D" should be replaced with "25 oh vitamin D". 

Author Response

To Editor,

We revised the manuscript with modifications and changes based on the new reviewer’s comments and editor’s comment (pdf notes).

We send you the revised manuscript together with our point-by-point response.

The changes in the text are written in red.

Thanking you in advance for your kind collaboration and suggestions.

Best regards,

The authors

Reviewer 2

This study analyzes and describes the physio-pathological interaction between bone mineral density, muscle mass and visceral adipose tissue (VAT) in old women using the structural equation modeling (SEM). This article is certainly original and interesting, in the light also of the recent literature on the metabolic effects of VAT. H

A: we are grateful to you for this valuable comment and appreciation and specially for the time that you invested reading our paper.

However, in my opinion, the analysis relating to bone tissue should be implemented. In the study, only "T score" femur and risk of hip fracture (FRAX) are described. I think that, the authors could complete the analysis by also adding: BMD and T score of femur neck, BMD and T score of lumbar spine (L1-L4), and risk of fractures at 10 years (FRAX) alo of major osteoporotic fractures. Moreover, are Bone turnover markers available? could they be associated to VAT?

A: Unfortunately, our database did not include the t score of femur neck. Regarding the t score of lumbar spine, a wide number of patients showed severe spinal / lumber compression. It was considered a strong bias we included this in the exclusion criteria.

We included this also in the limitation.

Bone turnover markers were not available.

 In table 1 and in the text: "vitamin D" should be replaced with "25 oh vitamin D". 

Corrected

Reviewer 3 Report

This study tried to discover the relation between visceral adipose tissue,  bone mineral density, and muscle mass using the structural equation modelling. 

#major

1. The authors tried to investigate the interaction of VAT, muscle mass and bone mineral density in elderly women by including muscle mass, bone mineral density, and visceral adipose tissue, as well as nutrition assessment,  cognition status, and various blood tests showing health status.

It is necessary to clearly define the variables used in this study in Table 1 and confirm their necessity. 

2. It is difficult to understand the background of this study because the contents are fragmentarily described in each part of the introduction and discussion. 

3. Some of the contents and tables seem to be described with inappropriate medical definitions. (ex. metabolic syndrome, azotemia). 

4. Appropriate names for variables in the table must be used. 

(ex. table 2 tscorefemore, fraxanca, azototemia..)

Author Response

To Editor,

We revised the manuscript with modifications and changes based on the new reviewer’s comments and editor’s comment (pdf notes).

We send you the revised manuscript together with our point-by-point response.

The changes in the text are written in red.

Thanking you in advance for your kind collaboration and suggestions.

Best regards,

The authors

Reviewer 3

It is necessary to clearly define the variables used in this study in Table 1 and confirm their necessity. 

Thanks a lot for all your valuable comments. We did our best to improve the paper. Table 1 has been described extensively also in according reviewer 1 indications.

  1. It is difficult to understand the background of this study because the contents are fragmentarily described in each part of the introduction and discussion. 

A: We improved the discussion including the major limitations and we evaluated the alignment between introduction and discussion focusing on the main outcomes under investigation.

  1. Some of the contents and tables seem to be described with inappropriate medical definitions. (ex. metabolic syndrome, azotemia). 4. Appropriate names for variables in the table must be used. 

(ex. table 2 tscorefemore, fraxanca, azototemia..)

A: We corrected all inappropriated definitions in table

Round 2

Reviewer 3 Report

1. Azotemia is a medical condition characterized by abnormally high levels of nitrogen-containing compounds. 

Does Azotemia mean Blood Urea Nitrogen in your Paper?

2. The paragraphs are divided so often that they are less readable. Please share the paragraphs for each theme.

Author Response

QUESTION

  1. Azotemia is a medical condition characterized by abnormally high levels of nitrogen-containing compounds. Does Azotemia mean Blood Urea Nitrogen in your Paper?

ANSWER:  Yes, Azotemia is blood urea nitrogen (BUN) in our paper. We clarified this meaning into the manuscript replacing or clarifying the term “azotemia” with “blood urea nitrogen” (BUN)

QUESTION

  1. The paragraphs are divided so often that they are less readable. Please share the paragraphs for each theme.

ANSWER:  thanks a lot for your suggestion. We divided the discussion following these specific sub-paragraphs:

  1. 1. The negative effects of VAT on muscular status and strenght
  2. 2. The Nutritional status’ mediation between visceral fat and muscle mass
  3. 3. Muscle deterioriation with age and comorbidities
  4. 4. Interaction between Muscle and Bone
  5. 5. The effects of cognitive stauts on muscle deterioration
  6. 6 Limitations of the study.